**Data Availability Statement:** All relevant data are within the manuscript and its Supporting information files.

# Modeling the initial phase of SARS-CoV-2 deposition in the respiratory tract mimicked by the $^{11}$C radionuclide

Heitor Evangelista[1]☯, César Amaral[1,2]☯, Luís Cristóvão Porto[3]‡, Sérgio J. Gonçalves Junior[1]*, Eduardo Delfino Sodré[1]☯, Juliana Nogueira[1]☯, Angela M. G. dos Santos[3]‡, Marcio Cataldo[4]☯, Daniel Junger[1]‡

1 LARAMG – Laboratory of Radioecology and Global Changes, Rio de Janeiro State University, Rio de Janeiro, Rio de Janeiro, Brazil, 2 LDD – Laboratorio de Diagnósticos por DNA, Rio de Janeiro State University, Rio de Janeiro, Rio de Janeiro, Brazil, 3 HLA – Laboratório de Histocompatibilidade e Criopreservação, Rio de Janeiro State University, Rio de Janeiro, Rio de Janeiro, Brazil, 4 Draxos Consultoria e Gestão Ambiental Ltda., Rio de Janeiro, Rio de Janeiro, Brazil

☯ These authors contributed equally to this work.
‡ These authors also contributed equally to this work.
* sjrgoncalves@gmail.com

## Abstract

The knowledge on the deposition and retention of the viral particle of SARS-CoV-2 in the respiratory tract during the very initial intake from the ambient air is of prime importance to understand the infectious process and COVID-19 initial symptoms. We propose to use a modified version of a widely tested lung deposition model developed by the ICRP, in the context of the ICRP Publication 66, that provides deposition patterns of microparticles in different lung compartments. In the model, we mimicked the "environmental decay" of the virus, determined by controlled experiments related to normal speeches, by the radionuclide $^{11}$C that presents comparable decay rates. Our results confirm clinical observations on the high virus retentions observed in the extrathoracic region and the lesser fraction on the alveolar section (in the order of 5), which may shed light on physiopathology of clinical events as well on the minimal inoculum required to establish infection.

## Introduction

Respiratory infections are relevant clinical conditions due to their diffusion, and potentially severe consequences, such as presently observed for the SARS-CoV spread all around the world. Declared as a "global pandemic" by the World Health Organization on March 11th, 2020, the new coronavirus disease 2019 (COVID-19) represents one of the most significant public health challenges in recent decades [1]. Respiratory infections caused by viruses, mainly due to their high capacity of infection and spread, are a major cause of illness and death from a worldwide perspective. In 2003, with the global alert caused by the SARS (Severe Acute Respiratory Syndrome), the coronaviruses joined this group, triggered by a newly identified coronavirus (SARS-CoV). Since then, several other pathogens associated with acute respiratory

**Funding:** The authors received no funding for this research.

**Competing interests:** We like to highlight that the commercial company Draxos Consultoria e Gestão Ambiental Ltda (by co-author Cataldo, M.) did not play a role in the study design, data collection, and analysis, decision to publish, or preparation of the manuscript and did not provide any financial support the research. This does not alter our adherence to PLOS ONE policies on sharing data.

system disorders have been described, such as more aggressive strains of Influenza, MERS-CoV in the Middle East, and also new types of coronavirus NL63 and KHU1 [2, 3]. Therefore, recognizing the modes of transmission of these emerging infectious diseases is a vital factor for both the safety of health staff, who interact with infected individuals, and the public, who will be exposed sooner or later to areas where these agents are circulating.

Airborne transmission is a key issue for the understanding of SARS-CoV-2 spread out, and it is particularly important for the community of healthcare professionals since they are more exposed to infected patients. However, the potential through contagion for the overall population related to the high agglomeration of citizens, as observed in various modes of urban transport in large cities, especially in developing countries, is not yet fully clarified. In Italy (Bergamo city), a first study showed that RNA SARS-CoV-2 might be present in association with microparticles in the outdoor environment. Still, the detection of the virus itself was mostly inconclusive [4]. While in the free atmosphere, virus particles undergo natural denaturation or inactivation conditions due to solar radiation, relative humidity, and air temperature [5–9]. This condition leads to the dehydration of virus particles caused by speech, sneezing, or coughing. As a result, the virus particles agglutinate with other organic and inorganic molecules/particles suspended in the air, and this interaction causes their size distribution to change. Therefore, this will affect its diffusion/dispersion/deposition and residence time in the atmosphere. Unlike other stochastic models (that is, the fraction of deposition in the lung compartment is determined based on the aerodynamic properties of aerosols), we here provide a simulation of a model that takes into account the full biokinetics of the radionuclide $^{11}$C mimicking particles containing viruses. To perform that, we used a modified version of a widely tested lung deposition model developed by the International Commission on Radiological Protection (ICRP), in the context of the ICRP Publication 66 (Human Respiratory Tract Model for Radiological Protection-66) from the ICRP Task Group on Internal Dosimetry.

## Materials and methods

During the last decades, several studies have deeply investigated the dynamics of respiratory infections to gain information on effective treatment/prevention of these clinical events [10]. Many different models have been developed till then. A respiratory infection model is a system that emulates the complexities observed on the relationship between the infectious agents and the host's defenses. Several *in vivo* and *in vitro* respiratory models for humans and other vertebrates are available and could be easily reproduced. Mathematical models are also available and were created to numerically describe the principles and evolution of respiratory infections and their diffusion. All of them require specific inputs and have complexities related to the condition they attempt to emulate, therefore presenting both advantages and limitations.

### The ICRP model

The ICRP (International Commission on Radiological Protection) has developed models for aerosol pulmonary deposition, given the intensive use of natural and artificial radionuclides in the nuclear industry. Such activities range from uranium mining, where workers are exposed to dust containing naturally occurring radioactive aerosols [11], to sectors where there is the handling of wastes from nuclear power plants and related facilities [12, 13]. Such models depend on the aerosol size distribution of the radioactive aerosols suspended in the air, their chemical form, and its solubility and the corresponding biokinetic processes associated [14]. In this context, extensive research on the biokinetics of radioactive aerosols has led to successful practical application and has been greatly improved in calculating the

internal dose of workers and people, especially those living in regions with high levels of natural radioactivity [15].

Herein we use a radionuclide-based model starting with the premise that once in the air the virus particles have an "equivalent" environmental half-life arising from their settling properties and interactions. Despite the human physiology and the biokinetics inherent in the ICRP model, one of its main parametrizations is related to the choice of the radioisotope to be used. From the experimental work performed by Stadnytskyi et al. [16], we draw an analogy between the exponential decrease of droplets nuclei in the air and a radionuclide decay by the following way: Assuming that each speech droplet nuclei have a viral load, the number of droplets caring virus particles vanish in the air, by gravimetric action, with half-life of 14 min, that is comparable to the disintegration rate of the nuclide 11C (half-life of 20.33 min). Droplets sedimentations half-life inferred by Stadnytskyi et al. [16], which used a highly sensitive laser light scattering system to track the dynamics of airborne speech droplets, is consistent with early studies of Knight (1973) [17] that estimated a value of 17 min. Thus, we can attribute to the virus an analog "constant of disintegration" related to the outdoor environment. Droplets were produced during the normal speech in indoor condition. Cough aerosols and of exhaled breath from patients or positive individuals tend to be similar in size distribution [18], with a predominance of pathogens in small particles, this is < 4–5 μm, with a median between 0.7 and 1.0 μm [19]. Since droplets nuclei have low densities, they may remain airborne for long time under most indoor conditions, unless there is removal due to natural or forced air currents [20]. Nevertheless, the virus load containing in aerosols will change in time since SARS-CoV viability decay significantly over a 3-h period, on basis of experimental aerosol generated in laboratory [21]. This provides enough condition to potential airborne source of virus transmission in low disturbed places. Also, the indoor environment is typically of lower relative humidity and viral particles tend to be most stable in such conditions. This in part explain why infections with lipid-enveloped viruses occurs most frequently during the winter season [22].

In order to evaluate the consistency of using the speech droplets vanishing pattern in time, as an analog to the [11]C decay (Fig 1a), we have conducted the Chi-Squared test using corresponding data for each 5 minutes interval. The result, with degree of freedom 16, was Chi-squared value 5.74 and p-value 0.0094, which means that no statistical difference exists between the two set of data at 0.05 confidence level.[11]C is one of the most useful radionuclides employed for Positron Emission Tomography (PET) radiochemistry because its attachment to a biologically active molecule does not modify the biochemical properties of the inoculated compound [23]. From the above, we elected the [11]C to mimic the virus intake and deposition in the respiratory tract. Another reason relies on the fact that carbon molecules are one of the base constituents of cells and tissues; this allows very precise information on metabolism processes, receptor/enzyme function, and biochemical mechanisms. As a complementary input to the model, we used airborne SARS-CoV-2 size distribution data, by Liu et al. [24], in a stand of theoretical curves or data derived from other virus types as influenza, Fig 1b. On that study, the size distribution of viral particles from a sequence of speeches in pre-sterilized gelatin filters installed inside a miniature cascade impactor during an air monitoring in the Renmin Hospital of Wuhan University—China, in February-March 2020 was obtained. As a result, they found that the aerodynamical diameter of particles varied from 5.0 to 5.5 μm. Other studies of droplets produced during speeches with sustained vocalization found modes from 1.8 to 5.5 μm [25].

The imperative assumption to allow the use of the ICRP model, which is specially designed for use by radionuclides, is to attribute a "decay property" to the virus. However, since the virus is no longer subject to any "decay" after it has been absorbed by the human organism, the assumption of mimicry by [11]C stops at that stage. The concentration of active virus particles

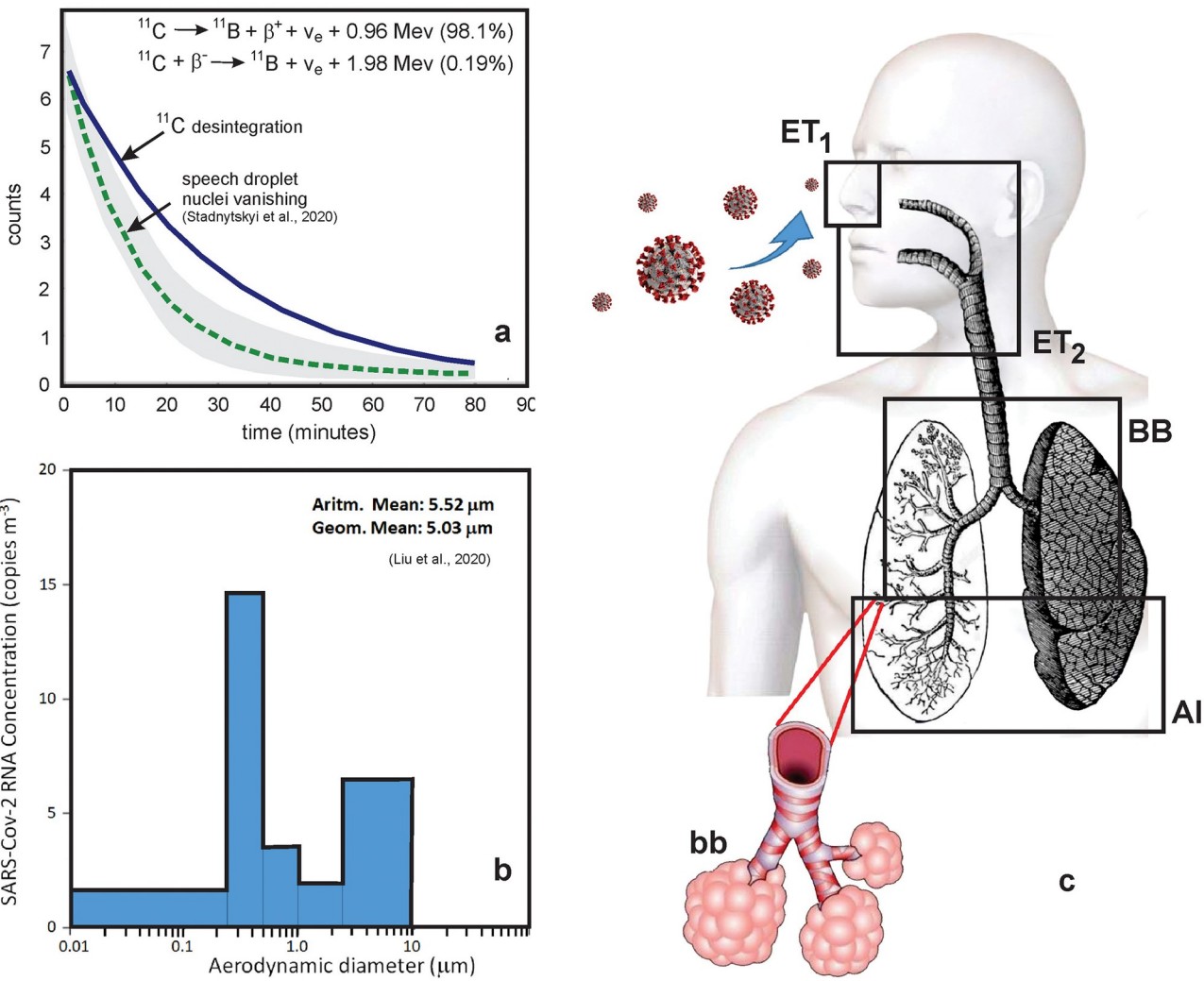

**Fig 1.** Basic inputs for the lung deposition model: (a) disintegration curve of the radionuclide [11]C (in bold) and speech droplet nuclei vanishing (dotted line) and variance (gray shaded area). See the data at the S1 Material; (b) size distribution of SARS-Cov-2 obtained by Liu et al., (2020) [24]; (c) respiratory tract compartments investigated by the model: AI (alveolar), bb (bronchioles), BB (bronchi), $ET_1$ (extrathoracic region 1—retention of material deposited in the anterior nose) and $ET_2$ (extrathoracic region 2 –retention of material deposited in the posterior nasal passage, larynx, pharynx, and mouth).

inoculated in the body remains the same until physiological reactions begin, and virus elimination/cell infections and replication occur. It, therefore, means that our estimates refer only to the initial deposition of the virus and that the deposition fractions presented here have been calculated for the first 1 minute after deposition.

The model used subdivides the respiratory system into two basic compartments: the extrathoracic region (composed of nose and trachea) and the thoracic region (composed of the bronchi to the alveolar sacs). The compartments AI, bb, BB, $ET_1$, and $ET_2$ represent the alveolar, bronchioles, bronchi, and extrathoracic regions, respectively, Fig 1c. The lung is considered to have 16 generations where bifurcations occur in the bronchial and bronchioles tree. The BB region comprises the generations from 0 to 8 and the bb from 9 to 15. The rates of mechanical removal of a compartment are expressed in 1/time. Ciliary transport and retention time in the tissues migrating to the pulmonary lymph nodes are also considered in the model,

**Table 1. Specifications of model parameters and inputs.**

| AIDE model parameters | AIDE model inputs |
|---|---|
| Model | C, Isotope: C-11 |
| Data source | Carbon according to ICRP-67* |
| Subject | Reference Worker (breathing rate of 1.2 m$^3$ h$^{-1}$) |
| Intake type | Inhalation, Single |
| Initial or Daily Activity | 1.000E+00 Bq |
| Inhalation type | Respiratory Tract model: ICRP-66 |
| Compound | Type F |
| GI Tract Absorption Factor | $f_1$ = 1.000E+00 |
| AMAD (μm) | 5.000E+00** |

* ICRP, 1993. Age-dependent Doses to Members of the Public from Intake of Radionuclides–Part 2 Ingestion Dose Coefficients. ICRP Publication 67. Ann. ICRP 23 (3–4)

** Based on the geometric mean of Fig 1b.

as well as blood absorption from these compartments, causing then a dispute between mechanical removal and blood absorption. The compartments above act as source compartments in the infection of the lung tissues. All the calculations were performed by the AIDE (Activity and Internal Dose Estimates) software [26]. The basic inputs of the model are presented in Table 1, where AMAD means "aerosol containing the viral load" median aerodynamic diameter, which implies that 50% of the viral load in the aerosol is associated with particles of aerodynamic diameter greater than the AMAD (used when deposition depends mainly on inertial impaction and sedimentation). At the same time, a compound refers to a material classified according to its rate of absorption from the respiratory tract to body fluids. In this case, deposited Type F materials are those that are readily absorbed into body fluids from the respiratory tract (Fast absorption), and $f_1$ is the fractional absorption in the gastrointestinal tract.

## SARS-CoV-2 field data

Considering the fact that COVID-19 is highly associated through the respiratory airways, we also present data of nasopharyngeal swabs (1490) and bronco-tracheal aspirate (4) COVID-19 RT-PCR tests performed at Pedro Ernesto University Hospital (HUPE), one of the specialized units for COVID-19 treatment in the Rio de Janeiro City/Brazil. We used the data to observe the evolution of the pandemics moments just before and after the mandatory use of personal safety equipment (masks) has been implemented by the local government. The database for patients with COVID-19 started on March 29th (13th epidemiological week) up to July 12th (28th epidemiological week), when 939 patients were hospitalized, and a total of 1,494 individuals were tested at the HUPE. Among the realized tests, all the four bronco-tracheal aspirates resulted in negative for SARS-CoV-2. The remaining test results were presented in Table 2.

## Ethics statement

The use of COVID-19 data was approved by the Ethics in Research Committee of the Pedro Ernesto University Hospital under the project "Epidemiological, laboratory, and clinical profile of the COVID-19 pandemic of patients treated at Rio de Janeiro State University (UERJ)" (CAAE: 30135320.0.0000.5259). All COVID-19 typing data were de-identified prior to the analysis and available for the authors uniquely as totals, as presented on Table 2.

**Table 2. The number of detected new and total tests at Pedro Ernesto University Hospital from epidemiological week 13 to 28.**

| | | Hospitalizations | | | Total | | |
|---|---|---|---|---|---|---|---|
| Epidemiological Week | Date | Tests | Detected | (%) | Tests | Detected | (%) |
| 13 | 29-mar | 11 | 2 | 18.2 | 11 | 2 | 18.2 |
| 14 | 5-abr | 22 | 7 | 31.8 | 24 | 8 | 33.3 |
| 15 | 12-abr | 40 | 16 | 40.0 | 48 | 19 | 39.6 |
| 16 | 19-abr | 46 | 40 | 87.0 | 58 | 50 | 86.2 |
| 17 | 26-abr | 42 | 36 | 85.7 | 47 | 40 | 85.1 |
| 18 | 3-mai | 86 | 52 | 60.5 | 112 | 63 | 56.3 |
| 19 | 10-mai | 93 | 55 | 59.1 | 142 | 74 | 52.1 |
| 20 | 17-mai | 56 | 31 | 55.4 | 108 | 57 | 52.8 |
| 21 | 24-mai | 53 | 27 | 50.9 | 91 | 35 | 38.5 |
| 22 | 31-mai | 106 | 46 | 43.4 | 179 | 74 | 41.3 |
| 23 | 7-jun | 74 | 21 | 28.4 | 117 | 29 | 24.8 |
| 24 | 14-jun | 72 | 19 | 26.4 | 119 | 30 | 25.2 |
| 25 | 21-jun | 54 | 8 | 14.8 | 117 | 17 | 14.5 |
| 26 | 28-jun | 59 | 10 | 16.9 | 126 | 26 | 20.6 |
| 27 | 5-jul | 55 | 4 | 7.3 | 91 | 7 | 7.7 |
| 28 | 12-jul | 70 | 5 | 7.1 | 104 | 9 | 8.7 |

## Results

Our results point to a far more relevant deposition fraction at the extrathoracic compartments of ET1 accounting to 47.4 and ET2 to 48.78 for the virus intake after 1 minute. In contrast, the sum for the bronchial and bronchioles compartments corresponded to 3.75 (Fig 2). All the data are available at S2 Material.

## Discussions

For a comparison with a Stochastic Lung Deposition Model, as proposed by Madas et al. [27], they found 61.8% of the total inhaled mass fraction to be at the upper airways, ~8.5% for the acinar airways and ~5.5% for the bronchial compartment corresponding to a single inhalation. The differences between the two models can be attributed to the fact that Madas et al. [27] have used a mass size distribution of particles from influenza with modal values between 2–3 µm, derived from coughing of patients, while we used a SARS-Cov-2 data in conditions of speeches with AMAD ~ 5 µm. Our results stress the impact of the upper airways in the initial airborne virus retention in the respiratory tract since the total extrathoracic compartment may retain more than 96% of the virus load. Both models point to at least ~4% of contribution to the most inner parts of the lung (bronchial and bronchioles). In the SARS-Cov-2 size distribution we have used, it is evident an existing viral load in the ultrafine particle size range (< 0.1 mm diameter). Their behavior differs from other groups of particles as the fine and coarse modes, since the virus attached to ultrafine particles may be deposit in the inner lung compartments by diffusion mechanisms. From controlled experiments, it is known that ultrafine particles peak deposition occurs in lung regions that encompass the transition zone between the conducting airways and the alveolar region [28]. Therefore, as our model predicts a small fraction of virus-containing particles reaching directly to the alveolar region, its significance in disease development should be considered. The SARS-Cov-2 can bind to the cells in that compartment via ACE2 (Angiotensin-Converting Enzyme 2), which are the host cell receptor responsible for mediating SARS-CoV-2 infection [29]. ACE2 plays an important role in the

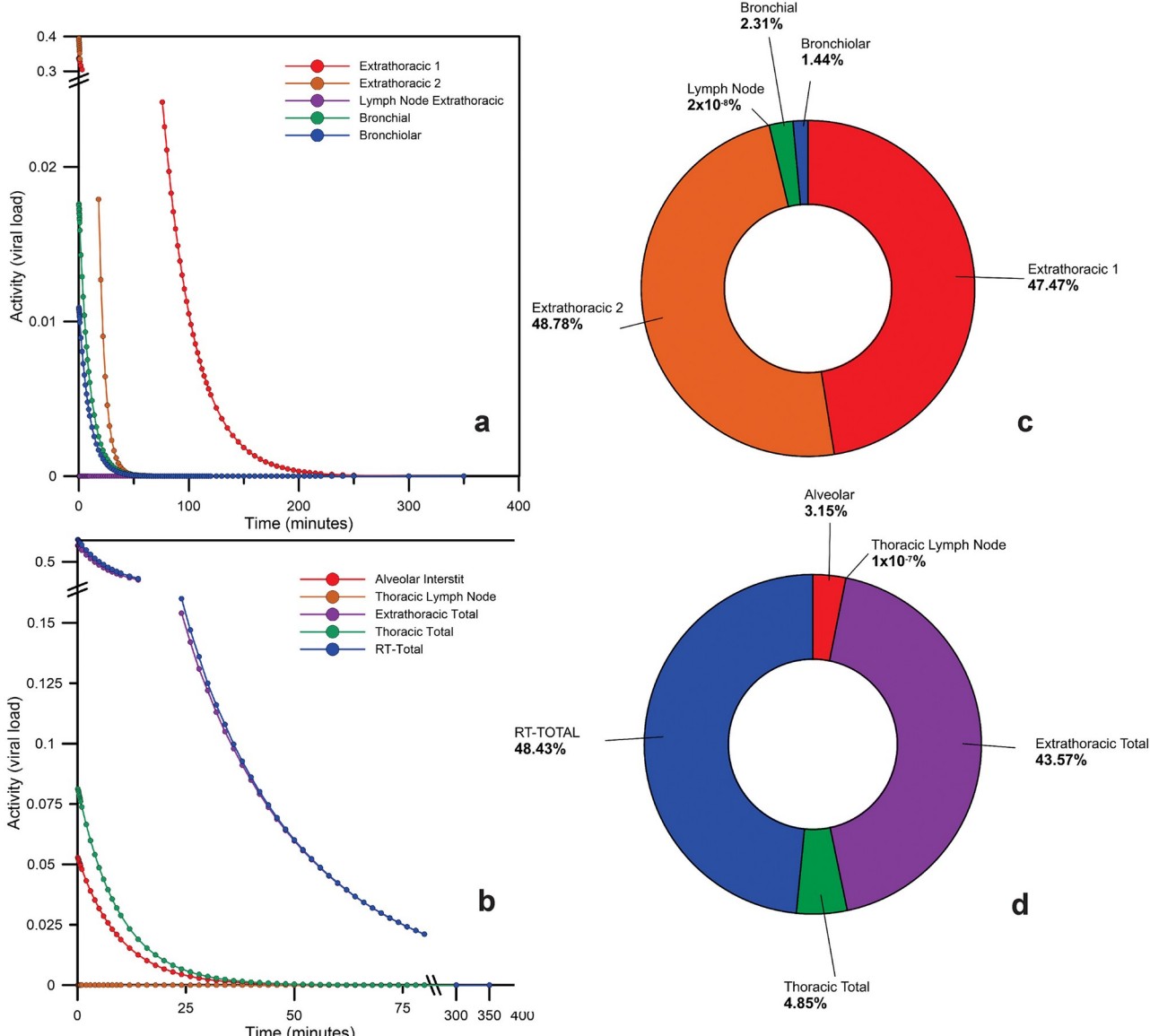

**Fig 2. Time-dependent particle transport from each respiratory tract region in the compartments of the model for SARS-Cov-2 (after 1 minute of exposure): (a) and (b) time response curves; (c) and (d) fractions of the initial intake of the virus by $ET_1$ pathway.**

lungs protecting it from acute respiratory distress syndrome (ARDS) by breaking down Angiotensin II, which has inflammatory effects [30, 31]. Binding of the SARS-CoV-2 to ACE2 inhibits it and thus weakens its protective action on the organ [32].

The pathogenesis and virus transmission pathways are still being investigated, and the already published results are, in fact, even under intense debate. That is the case for the airborne transmission of the virus. The analyses of the temporal dynamics of the SARS-Cov-2 infection indicates that the viral shedding begins after 2–3 days of the first symptoms, thereby promoting pre-symptomatic/asymptomatic transmission of the virus [33]. In this sense, understanding the first phases of the virus infection is paramount. Our results were derived from normal speech conditions as viral source-term and indoor environments. In this specific case, we observed extrathoracic percentage infection levels in good agreement with clinical

observations of patients that initially presented mild COVID-19 symptoms evolving to a more deteriorated health board [34]. Though COVID-19 manifestation linked to a minimal infectious dosage, as expected in the alveolar compartment, is not yet clear, given that: (1) no masks have a 100% retention efficiency; (2) SARS-CoV-2 viral particles are found in aerodynamic diameters shorter than 0.1 μm in surveys; (3) the model predicts around 5% direct penetration of viruses in the alveolar compartments. These facts, when combined, may explain why a fraction of the population, even using masks, get contaminated with "no apparent reason." Respiratory tract models developed by the ICRP Task Groups have been largely used for safety and protection in nuclear activities in several countries, reaching excellent performance and being validated by internal measurements. Their use for non-radioactive aerosols of biological and mineral origin and pollutants is an emerging topic and a potential to be explored.

Since ET1 and ET2 are key-compartments in the initial phase of SARS-Cov-2 deposition, we investigated the impact of the use of individual protection such as masks and face shields on the epidemiological data related to the COVID-19. Table 2 shows a survey developed by our working group from nasopharyngeal swabs and bronco-tracheal aspirate tests for COVID-19 by RT-PCR performed at Pedro Ernesto University Hospital (HUPE), in Rio de Janeiro, Brazil. The city of Rio de Janeiro in no time had adopted a full lockdown strategy, but just recommendations of social distance and personal care such as hands washing and alcohol use for individual and items/objects disinfection. However, COVID-19 data showed continuous prevalence forcing the local authorities to carry out a mandatory use of masks on April 23rd.

Our results suggest that the fraction of both hospitalized and total tested patients with SARS-CoV-2 detected from the RT-PCR tests exhibited a significant decrease when we observed an immediate drop in percentage from 85% to 60% (Fig 3). We do believe that the

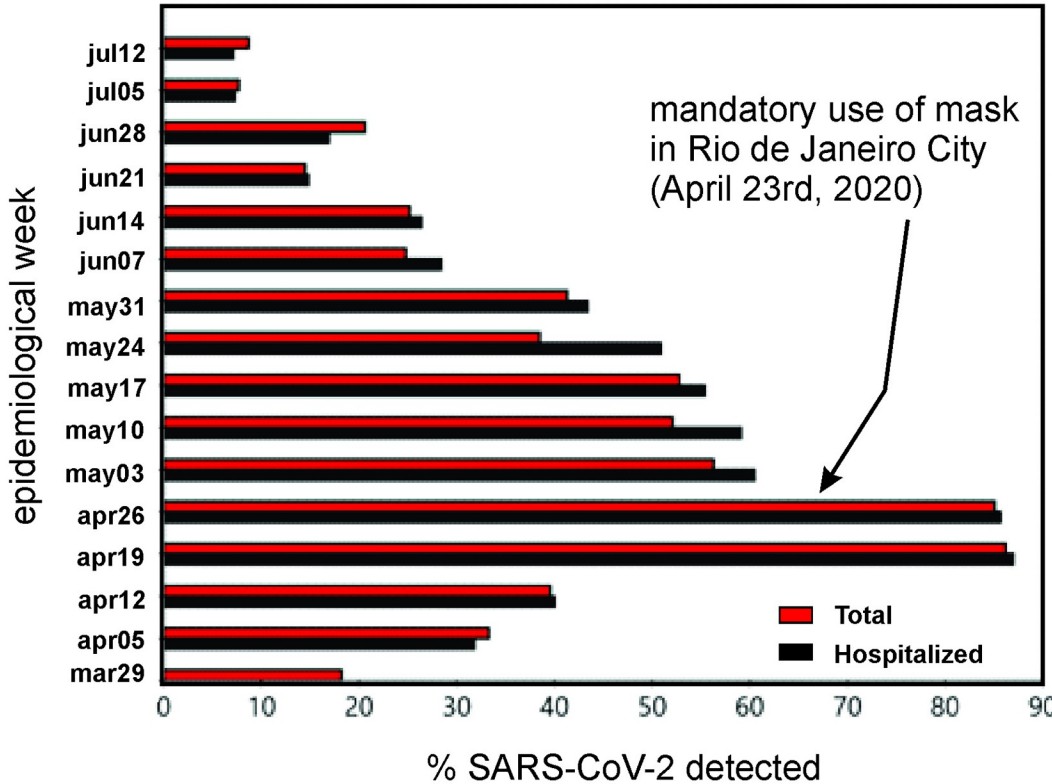

**Fig 3. SARS-CoV-2 detection from nasopharynx COVID-19 RT-PCR tests performed at Pedro Ernesto University Hospital (HUPE) in Rio de Janeiro/Brazil.**

analyzed sample is representative of the city population since the HUPE receives patients from several health units spread all over the urban domain. We should also note that due to socio-economic reasons and also difficulties in the acquisition of high-quality safety equipment, most of the population made use of simple, cheap, and home-made masks and face shields. However, despite these circumstances, effective response against the pandemic was achieved.

A measure of the viral load that is initially concentrated at the extrathoracic compartments $ET_1$ and $ET_2$ is provided by the nasopharyngeal swab/culture method, a clinical test sample of nasal secretions from the back of the nose and throat. Ct (Cycle threshold) values in a real time PCR assay display the viral load, in a way that Ct levels are inversely proportional to the amount of target nucleic acid in the sample. Our observation, in the same Hospital complex, on the temporal evolution of the Ct values, since the very start of the pandemic in Rio de Janeiro City, depicted a very surprising picture, in which the Ct values tended to higher values from March to September 2020 (Fig 4). In average it changed from 28 to 35 that represents a considerable increase.

Previous studies using chest computed tomography (CT) of SARS CoV-2 RT-PCR positive patients showed that a statistically correlation exist between Ct values and the total severity score (TSS) of acute lung inflammation derived from CT [35]. Ct values are normally categorized as high <20, 20 < Ct < 29 are strong positive, 30 < Ct < 37 are moderate positive, 38 < Ct < 40 are indicative of minimal amounts of target nucleic acid, and very low loads range between 40 and 45. According to our Ct data trend and the TSS categorizations, our Ct varied from strong positive to moderate positive while TSS corresponded to moderate/severe to mild lung inflammation. The running average of Covid-19 death per day in the city population followed that trend of lowering fatalities from autumn to winter season (Fig 4). We believe that the nature of this behavior could be related with several causes such as changes in aerosol dynamics between summer/autumn to winter season, the more extensive use of masks, the late

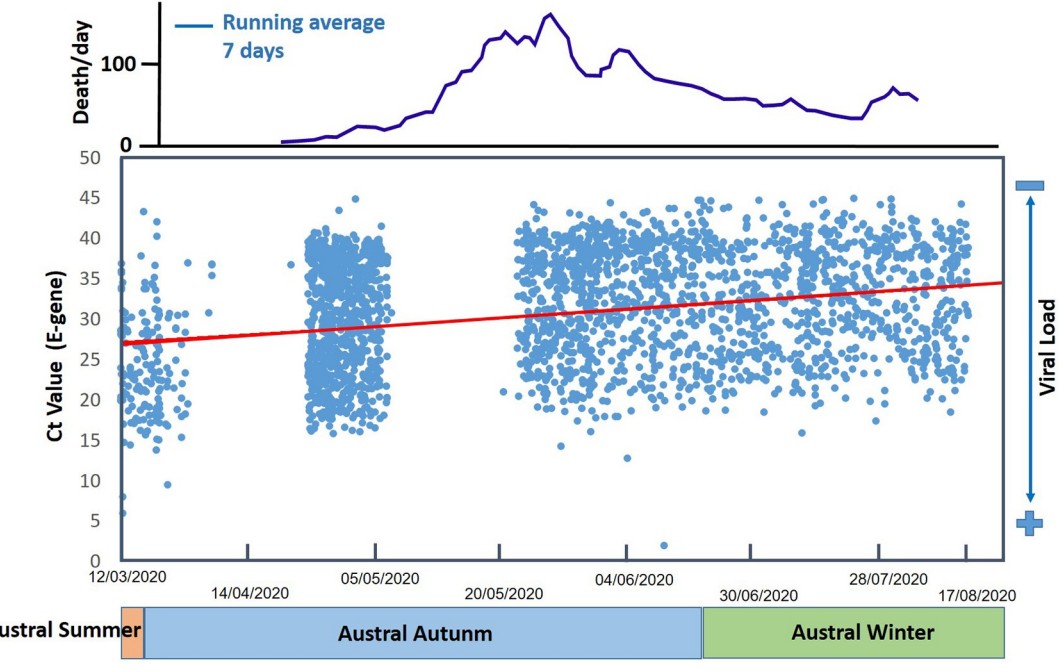

**Fig 4. Seasonal behavior of Ct in a hospital of Rio de Janeiro City and Covid-19 daily mortality for the city population.**
(n = 15251, S2 Material).

notification of the disease since in Brazil there is no effective politics for massive Covid-19 *testing*, and also the prevalence of other infectious SARS-CoV-2 strains, as pointed by Korber et al. (2020) [36]. *For a Covid-19 second wave to come (predicted to austral summer of 2020/2021), it is still inconclusive if Ct values does represent a linear trend of lowering or it represents a seasonal aspect of Covid-19 considering the environmental factors involved and the local social behaviors.*

In summary, the respiratory tract models produced by IAEA Task Group have been widely used in many countries for health and defense in nuclear activities, reaching excellent performance and being validated by *in locus* measurements. Their use for non-radioactive aerosols and microparticles of biological and mineral origin and pollutants is an emerging topic and a potential to be explored. It may also be useful as a tool to design strategies based on risk stratification considering the global public health emergency by COVID-19 pandemics.

## Supporting information

**S1 Material. Data for Fig 1, 11C Decay and droplets decay.**
(XLSX)

**S2 Material. Data for Fig 2, deposition fraction at the lung's compartments.**
(XLSX)

**S3 Material. Epidemiological data for Fig 4.**
(XLS)

## Acknowledgments

We greatly thank Dr. Luiz Bertelli for the availability of AIDE (Activity and Internal Dose Estimates) software and important comments on the numeric model and Roberta Priori for draw designs.

## Author Contributions

**Conceptualization:** Heitor Evangelista, Daniel Junger.

**Data curation:** César Amaral, Luís Cristóvão Porto.

**Formal analysis:** César Amaral, Luís Cristóvão Porto, Angela M. G. dos Santos.

**Investigation:** Heitor Evangelista, César Amaral.

**Methodology:** Heitor Evangelista, César Amaral, Luís Cristóvão Porto, Angela M. G. dos Santos.

**Resources:** Heitor Evangelista.

**Supervision:** Heitor Evangelista, César Amaral.

**Validation:** Daniel Junger.

**Writing – original draft:** Heitor Evangelista, César Amaral, Sérgio J. Gonçalves Junior, Eduardo Delfino Sodré, Juliana Nogueira, Marcio Cataldo, Daniel Junger.

**Writing – review & editing:** Heitor Evangelista, César Amaral, Sérgio J. Gonçalves Junior, Eduardo Delfino Sodré, Juliana Nogueira, Marcio Cataldo, Daniel Junger.

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
