## [Decision Letter · Decision Letter 0]

23 Oct 2020

PONE-D-20-25540

Modeling the initial phase of SARS-CoV-2 deposition in the respiratory tract mimicked by the 11C radionuclide

PLOS ONE

Dear Dr. Goncalves Jr.,

Thank you for submitting your manuscript to PLOS ONE. After careful consideration, we feel that it has merit but does not fully meet PLOS ONE’s publication criteria as it currently stands. Therefore, we invite you to submit a revised version of the manuscript that addresses the points raised during the review process.

We look forward to receiving your revised manuscript.

Kind regards,

Simone Lolli

Academic Editor

PLOS ONE

Journal Requirements:

2. Please amend the Methods section of your manuscript to include the information about ethics approval, and data de-identification that you provided in the Ethics Statement.

'The authors received no funding for this research.The funders had no role in study design, data collection and analysis, decision to publish, or preparation of the manuscript.'

We note that one or more of the authors are employed by a commercial company: Draxos Consultoria e Gestão Ambiental Ltda.

Reviewers' comments:

Reviewer's Responses to Questions

**Comments to the Author**

1. Is the manuscript technically sound, and do the data support the conclusions?

Reviewer #1: Yes

Reviewer #2: Yes

Reviewer #3: Yes

2. Has the statistical analysis been performed appropriately and rigorously? 

Reviewer #1: Yes

Reviewer #2: Yes

Reviewer #3: I Don't Know

3. Have the authors made all data underlying the findings in their manuscript fully available?

Reviewer #1: Yes

Reviewer #2: No

Reviewer #3: Yes

4. Is the manuscript presented in an intelligible fashion and written in standard English?

Reviewer #1: Yes

Reviewer #2: Yes

Reviewer #3: Yes

5. Review Comments to the Author

Reviewer #1: The paper is well written and its topic is proper to the journal scope. My only concern regrads the clinical implication of the reported finding in clinical setting. I suggest to clearly discuss this critical issue at least in the discussion section of the manuscript

Reviewer #2: Review of PONE-D-20-25540 Modeling the initial phase of SARS-CoV-2 deposition in the respiratory tract mimicked by the 11C radionuclide by Dr. Sérgio José Goncalves Jr. The manuscript studies the initial phase deposition of SARS-Cov-2 in the respiratory tract mimicked by the 11 C radionuclide. The paper itself could make sense, but the part concerning the considered assumption should be explained much more in detail.

My main concern regards indeed the main assumption. Really the 11 C radionuclide behavior can be considered "similar" to a virus particle? To which extent? The narrative should be broaden and more convincing.

Some references were suggested (maybe the manuscript was submitted earlier)

The specific comments can be found in the attached manuscript

Reviewer #3: Very interesting study which may help in better understand pathogenesis of the disease and maybe why positif but asympthomatic covid patients can be contagious: a so high distribution of the virus in extrathoracic airways can explain how easy can be the spreading of droplets just by talking, breathing or touching the nose even in absence of cough.

6. PLOS authors have the option to publish the peer review history of their article (what does this mean?). If published, this will include your full peer review and any attached files.

Reviewer #1: **Yes: **Giulia Maria Stella

Reviewer #2: No

Reviewer #3: **Yes: **Claudia Collu

---

## [Author Response · Author response to Decision Letter 0]

15 Dec 2020

Dear Simone Lolli,

Academic Editor

PLOS ONE

We would like to thank the reviewers for the careful and thorough reading of this manuscript and the thoughtful comments and constructive suggestions, which help improve this manuscript's quality. All comments and suggestions have been appropriately addressed. The requested modifications/corrections were inserted directly into the manuscript (significant changes are highlighted in red). The answers to reviewers’ point-by-point are described below (the modifications are indicated by page and line number of Manuscript with changes marked in red). Also, we would like to add to our list of authors Dr. Daniel Junger because he contributed at medical survey and discussion, being an essential component for this work. We attached the form “request for change to authorship” properly filled. 

Yours sincerely,

Sérgio J. Gonçalves Jr (Corresponding author) on behalf Heitor Evangelista and co-authors

Reviewer #1: The paper is well written and its topic is proper to the journal scope. My only concern regards the clinical implication of the reported finding in clinical setting. I suggest to clearly discuss this critical issue at least in the discussion section of the manuscript

Authors answers: We have added the following paragraph in the discussion section in Pag 7/Line 252:

“A measure of the viral load that is initially concentrated at the extrathoracic compartments ET1 and ET2 is provided by the nasopharyngeal swab/culture method, a clinical test sample of nasal secretions from the back of the nose and throat. Ct (Cycle threshold) values in a real time PCR assay display the viral load, in a way that Ct levels are inversely proportional to the amount of target nucleic acid in the sample. Our observation, in the same Hospital complex, on the temporal evolution of the Ct values, since the very start of the pandemic in Rio de Janeiro City, depicted a very surprising picture, in which the Ct values tended to higher values from March to September 2020 (Figure 4). In average it changed from 28 to 35 that represents a considerable increase. Previous studies using chest computed tomography (CT) of SARS CoV-2 RT-PCR positive patients showed that a statistically correlation exist between Ct values and the total severity score (TSS) of acute lung inflammation derived from CT (Yagcia et al., 2020). Ct values are normally categorized as high <20, 20 < Ct < 29 are strong positive, 30 < Ct < 37 are moderate positive, 38 < Ct < 40 are indicative of minimal amounts of target nucleic acid, and very low loads range between 40 and 45. According to our Ct data trend and the TSS categorizations, our Ct varied from strong positive to moderate positive while TSS corresponded to moderate/severe to mild lung inflammation. The running average of Covid-19 death per day in the city population followed that trend of lowering fatalities from autumn to winter season (Figure 4). We believe that the nature of this behavior could be related with several causes such as changes in aerosol dynamics between summer/autumn to winter season, the more extensive use of masks, the late notification of the disease since in Brazil there is no effective politics for massive Covid-19 testing, and also the prevalence of other infectious SARS-CoV-2 strains, as pointed by Korber et al. (2020). For a Covid-19 second wave to come (predicted to austral summer of 2020/2021), it is still inconclusive if Ct values does represent a linear trend of lowering or it represents a seasonal aspect of Covid-19 considering the environmental factors involved and the local social behaviors.”

Fig 4. Seasonal behavior of Ct in a hospital of Rio de Janeiro City and Covid-19 daily mortality for the city population.

Also, we add some new references: 

Yagci, A. K., Sarinoglu, R. C., Bilgin, H., Yanılmaz, Ö., Sayın, E., Deniz, G., Guncu, M.M., Doyuk, Z., Can, B., Kuzan, B.N., Aslan, B., Korten, V., Cimsit, C. (2020). Relationship of the cycle threshold values of SARS-CoV-2 polymerase chain reaction and total severity score of computerized tomography in patients with COVID 19. International Journal of Infectious Diseases, 101, 160-166. doi:10.1016/j.ijid.2020.09.1449 

Korber B, Fischer WM, Gnanakaran S et al. (2020) Tracking changes in SARS-CoV-2 Spike: evidence that D614G increases infectivity of the COVID-19 virus. Cell doi:10.1016/j.cell.2020.06.043

Reviewer #2: Review of PONE-D-20-25540 Modeling the initial phase of SARS-CoV-2 deposition in the respiratory tract mimicked by the 11C radionuclide by Dr. Sérgio José Goncalves Jr. The manuscript studies the initial phase deposition of SARS-Cov-2 in the respiratory tract mimicked by the 11C radionuclide. The paper itself could make sense, but the part concerning the considered assumption should be explained much more in detail.

My main concern regards indeed the main assumption. Really the 11C radionuclide behavior can be considered "similar" to a virus particle? To which extent? The narrative should be broaden and more convincing.

Some references were suggested (maybe the manuscript was submitted earlier). The specific comments can be found in the attached manuscript

Authors answers:

Pag 1/Line 19: We have removed the “To give some light on that, ...”.

Pag 2/Line 50: We have added the recommended references to the list: 

Casanova, L. M., Rutala, J. S., Weber, W. A. & Sobsey, M. D. Effects of air temperature and relative humidity on coronavirus survival on surfaces. Appl. Environ. Microbiol. 76, 2712–2717 (2020).

Lolli, Simone, Ying-Chieh Chen, Sheng-Hsiang Wang, and Gemine Vivone. "Impact of meteorological conditions and air pollution on COVID-19 pandemic transmission in Italy." Scientific reports 10, no. 1 (2020): 1-15.

Sobral, M. F. F., Duarte, G. B., Sobral, A. I. G. D. P., Marinho, M. L. M. & Melo, A. D. S. Association between climate variables and global transmission oF SARS-CoV-2. Sci. Total Environ https://doi.org/10.1016/j.scitotenv.2020.138997 (2020).

Pag 2/Line 72: We changed “ICRP” to “ICRP (International Commission on Radiological Protection)”.

Pag 3/Lines 82-84: We better explained our assumption by changing the sentence: 

Original: “Herein we use a radionuclide-based model starting with the premise that the virus (SARS-Cov-2) has an "environmental half-life" due to its degradation process in the environment before the deposition in the respiratory tract. From this assumption, we draw an analogy with the half-life of a radionuclide”

to

Revised:

Pag 3/Line 85-95: “Herein we use a radionuclide-based model starting with the premise that once in the air the virus particles have an “equivalent” environmental half-life arising from their settling properties and interactions. Despite the human physiology and the biokinetics inherent in the ICRP model, one of its main parametrizations is related to the choice of the radioisotope to be used. From the experimental work performed by Stadnytskyi et al. [13], we draw an analogy between the exponential decrease of droplets nuclei in the air and a radionuclide decay by the following way: Assuming that each speech droplet nuclei have a viral load, the number of droplets caring virus particles vanish in the air, by gravimetric action, with half-life of 14 min, that is comparable to the disintegration rate of the nuclide 11C (half-life of 20.33 min). Droplets sedimentations half-life inferred by Stadnytskyi et al. [13], which used a highly sensitive laser light scattering system to track the dynamics of airborne speech droplets, is consistent with early studies of Knight (1973) that estimated a value of 17 min.”

From the above, we removed the sentence: “The choice of the radionuclide was made based on studies by Stadnytskyi et al. [13], which used a highly sensitive laser light scattering system to track the dynamics of airborne speech droplets generated by carriers with SARS-CoV-2” in Pag 3/Lines 97-99.

In Pag 3/Line 114 We have included the following sentence: 

“In order to evaluate the consistency of using the speech droplets vanishing pattern in time, as an analog to the 11C decay, we have conducted the Chi-Squared test using corresponding data for each 5 minutes interval. The result, with degree of freedom 16, was Chi-squared value 5.74 and p-value 0.0094, which means that no statistical difference exists between the two set of data at 0.05 confidence level.”

New references: 

Knight, V. 1973. Airborne transmission and pulmonary deposition of respiratory viruses. In Viral and Mycoplasma Infections of the Respiratory Tract. V. Knight, Ed.: 1-9. Lea & Febiger. Philadelphia.” 

Pag 3/Lines 99-101: We have specified better the following sentence: ‘’Droplets were produced during the normal speaking condition, and can be suspended in the air for a tenth of minutes and, therefore, can be a potential source of airborne virus transmission in low disturbed places”

Revised: 

Pag 3/Lines 101-111: “Droplets were produced during the normal speech in indoor condition. Cough aerosols and of exhaled breath from patients or positive individuals tend to be similar in size distributions (Fennelly, 2020), with a predominance of pathogens in small particles, this is < 4-5 µm, with a median between 0.7 and 1.0 µm (Bake et al., 2019). Since droplets nuclei have low densities, they may remain airborne for long time under most indoor conditions, unless there is removal due to natural or forced air currents (Wells, 1955). Nevertheless, the virus load containing in aerosols will change in time since SARS-CoV viability decay significantly over a 3-h period, on basis of experimental aerosol generated in laboratory (Van Doremalen et al., 2020). This provides enough condition to potential airborne source of virus transmission in low disturbed places. Also, the indoor environment is typically of lower relative humidity and viral particles tend to be most stable in such conditions. This in part explain why infections with lipid-enveloped viruses occurs most frequently during the winter season (Knight, 1980). 

New references: 

Fennelly, K. P. (2020). Particle sizes of infectious aerosols: implications for infection control. The Lancet Respiratory Medicine. doi:10.1016/s2213-2600(20)30323-4 

Wells WF. Aerodynamics of droplet nuclei. In: Airborne contagion and air hygiene: an ecological study of droplet infections. Cambridge: Harvard University Press, 1955: 13–19.

Bake B, Larsson P, Ljungkvist G, Ljungström E, Olin AC. Exhaled particles and small airways. Respir Res 2019; 20: 8

Van Doremalen N, Bushmaker T, Morris DH, et al. Aerosol and surface stability of SARS-CoV-2 as compared with SARS-CoV-1. N Engl J Med 2020; 382: 1564–67

Knight, V. (1980). Virus as agents of airborne contagion. Annals New York Academy of Sciences. Part V: 147-156.

From the above, we have removed the following sentence in Pag 3/Lines: 112-114:

“The settling down (vanishing) time-scale of droplets before dehydration and attachment to other existing microparticles in the atmosphere describes a behavior similar to the artificial radioisotope 11C disintegration curve, which is 20.33 minutes in half-life (Fig.1a)”

Pag 3/Lines 130-131: “Other studies of droplets produced during speeches with sustained vocalization found modes from 1.8 to 5.5 �m [16].”

Authors answers: We have included more details above.

Reviewer #3: Very interesting study which may help in better understand pathogenesis of the disease and maybe why positif but asympthomatic covid patients can be contagious: a so high distribution of the virus in extrathoracic airways can explain how easy can be the spreading of droplets just by talking, breathing or touching the nose even in absence of cough.

Authors answers: Nothing to reply.

#Additional minor changes by the authors:

Pag 1/Lines 25-27: We have changed the sentence in the Abstract: “which relevance is a subject to be investigated” to “which may shed light on physiopathology of clinical events as well on the minimal inoculum required to establish infection.”

Pag 7/Line 281-283: We have added the sentence: “It may also be useful as a tool to design strategies based on risk stratification considering the global public health emergency by COVID-19 pandemics.”

Pag 6/Line 184: We amend the Methods section to include the information about ethics approval and data de-identification that we provided in the Ethics Statement.

Pag 4/Line 133: We add the Supplementary Information for Fig 1.

Pag 6/Line 183: We add the Supplementary Information for Fig 2.

Pag 8/Line 284: We add the Supplementary Information for Fig 4.

Pag 12/Line 410: We add the Supplementary Information captions.

---

## [Decision Letter · Decision Letter 1]

21 Dec 2020

Modeling the initial phase of SARS-CoV-2 deposition in the respiratory tract mimicked by the 11C radionuclide

PONE-D-20-25540R1

Dear Dr. Goncalves Jr.,

We’re pleased to inform you that your manuscript has been judged scientifically suitable for publication and will be formally accepted for publication once it meets all outstanding technical requirements.

Kind regards,

Simone Lolli

Academic Editor

PLOS ONE

Additional Editor Comments (optional):

The authors resolved all the previously raised issues and now the manuscript is ready for publication.

Reviewers' comments:

Reviewer's Responses to Questions

**Comments to the Author**

1. If the authors have adequately addressed your comments raised in a previous round of review and you feel that this manuscript is now acceptable for publication, you may indicate that here to bypass the “Comments to the Author” section, enter your conflict of interest statement in the “Confidential to Editor” section, and submit your "Accept" recommendation.

Reviewer #2: All comments have been addressed

2. Is the manuscript technically sound, and do the data support the conclusions?

Reviewer #2: Yes

3. Has the statistical analysis been performed appropriately and rigorously? 

Reviewer #2: Yes

4. Have the authors made all data underlying the findings in their manuscript fully available?

Reviewer #2: Yes

5. Is the manuscript presented in an intelligible fashion and written in standard English?

Reviewer #2: Yes

6. Review Comments to the Author

Reviewer #2: The authors addressed all my previously raised issues and the manuscript is ready for publication after the correction of some typos.

7. PLOS authors have the option to publish the peer review history of their article (what does this mean?). If published, this will include your full peer review and any attached files.

Reviewer #2: No

---

## [Editor Report · Acceptance letter]

28 Dec 2020

PONE-D-20-25540R1 

Modeling the initial phase of SARS-COv-2 deposition in the respiratory tract mimicked by the _11_C radionuclide 

Dear Dr. Gonçalves Junior:

I'm pleased to inform you that your manuscript has been deemed suitable for publication in PLOS ONE. Congratulations! Your manuscript is now with our production department. 

Kind regards, 

on behalf of

Dr. Simone Lolli 

Academic Editor

PLOS ONE